# Concentric hollow multi-hexagonal platelets from a small molecule

Chenglong Liao[1,2,6], Yanjun Gong[1,2,6], Yanxue Che[3], Hongwei Ji[1,2], Bing Liu [2,4] ✉, Ling Zang [5] ✉, Yanke Che [1,2] ✉ & Jincai Zhao [1,2]

The creation of well-defined hollow two-dimensional structures from small organic molecules, particularly those with controlled widths and numbers of segments, remains a formidable challenge. Here we report the fabrication of the well-defined concentric hollow two-dimensional platelets with pro-grammable widths and numbers of segments through constructing a con-centric multiblock two-dimensional precursor followed by post-processing. The fabrication of concentric multi-hexagons two-dimensional platelets is realized by the alternative heteroepitaxial growth of two donor-acceptor molecules. Upon ultraviolet irradiation, one of the two donor-acceptor molecules can be selectively oxidized by singlet oxygen generated during the process, and the oxidized product becomes more soluble due to increased polarity. This allows for selective removal of the oxidized seg-ments simply by solvent dissolution, yielding hollow multiblock two-dimensional structures. The hollow two-dimensional platelets can be uti-lized as templates to lithograph complex electrodes with precisely con-trolled gap sizes, thereby offering a platform for examining the optoelectronic performance of functional materials.

Two-dimensional (2D) nanomaterials of semiconducting organic small molecules have attracting significant interests as they com-bined the performance from both semiconducting materials and 2D materials[1-13]. To modulate the properties of the 2D nanomaterials usually require the ability of precisely controlling their sizes, dimensions and especially their internal structures. For example, 2D platelets of block copolymers with compositionally distinct seg-mented cores that allow for selective degradation toward pro-grammed drug release and cargo delivery were designed and implemented by a seeded growth process[1]. Achieving such complex but hollow 2D nanostructures typically requires two steps. In the first step non-hollow 2D nanostructures was fabricated and in the second step some a segmented component was selectively removed. The first step can be fairly addressed by seeded heteroepitaxy growth techniques that has been used for the construction of complex multi-component solid 2D nanomaterials[14-29], however, spatially selective removal of segmented components remains a challen-ging task.

To date, spatially selective removal of segmented components in 2D nanostructures has only been reported in a few polymer-based examples, which inevitably require a complicated route by cross-linking one part and dissolving the non-crosslinked part[3,9]. Utilizing seeded heteroepitaxial growth to grow easily removed segments is an alternative method that has recently been successfully used for 2D materials of block copolymers[1]. However, spatially selective removal of segmented components for 2D monocrystalline nanostructures of small molecules has not been reported because such 2D nanos-tructures are difficult to achieve and, more importantly, there is a lack

[1]Key Laboratory of Photochemistry, CAS Research/Education Center for Excellence in Molecular Sciences, Institute of Chemistry, Chinese Academy of Sciences, Beijing, China. [2]University of Chinese Academy of Sciences, Beijing, China. [3]HT-NOVA Co. Ltd., Zhuyuan Road, Shunyi District, Beijing, China. [4]Beijing National Laboratory for Molecular Sciences, State Key Laboratory of Polymer Physics and Chemistry, Institute of Chemistry, Chinese Academy of Sciences, Beijing, China. [5]Department of Materials Science and Engineering, Nano Institute of Utah, University of Utah, Salt Lake City, UT, USA. [6]These authors contributed equally: Chenglong Liao, Yanjun Gong. ✉e-mail: liubing@iccas.ac.cn; lzang@eng.utah.edu; ykche@iccas.ac.cn

of an effective technique to remove the desired segments without destroying the materials.

Here we report the self-assembly of segmented hexagonal 2D platelets by seeded heteroepitaxial growth using donor-acceptor (D-A) molecules **1** and **2** (Fig. 1a) and demonstrate that the segments of **1** can be easily removed by selective photooxidation and solvent. Extensive experimental characterizations show that ultraviolet (UV) irradiation of segment **1** generates significantly more singlet oxygen than UV irradiation of segment **2**, thereby influencing the selectivity of photooxidation. The increased difference in polarity after irradiation allows a polar solvent to dissolve the oxidized molecule **1** while preserving the shape of molecule **2**. Well-defined concentric hollow 2D platelets can thus be achieved. Furthermore, this method is capable of creating hollow multi-hexagonal 2D platelets with programable widths and numbers of segments, which can be further used as templates to produce complex electrodes with a precision of about 200 nm by photolithography. The resulting electrodes should be very useful for investigating the optoelectronic properties and performance of functional materials.

## Results

### Molecule design and preparation of multi-hexagonal platelets

Multi-block 2D heteroplatelets were obtained by alternatively adding two different D-A molecules as building blocks, which contain fluorene and benzimidazole groups as the D part, and benzothiadiazole or benzoselenadiazole group as the A part[19]. The minor differences in molecular structure allow for the formation of heteroepitaxial structures through the continuous hydrogen bonds with a coformer (alcohol). While crucial for the fabrication of 2D structures, the enduring hydrogen bonding with alcohol in these arrangements obstructs the creation of hollow structures utilizing selective post-processing techniques. To overcome this limitation, we synthesized D-A molecule **1** in the current investigation (see detailed characterizations, Supplementary Figs. 5 and 6), which features a benzoselenadiazole group as the A part, but lacks hydrogen-bond forming groups (Fig. 1a). We envisioned that when assembled with molecule **2**[30], it could form concentric segmented 2D structures. Furthermore, in contrast to the benzothiadiazole group in **2**, the benzoselenadiazole group in **1**, is anticipated to boost intersystem crossing (ISC) to create triplet state via the heavy atom effect. Consequently, this process is

expected to produce singlet oxygen, leading to the oxidation of fluorene units in **1** and thereby imparting unique characteristics for subsequent post-processing, such as the ability to dissolve into hollow structures by a selected solvent. By employing a supersaturated solution-fostered seeded growth method[30], we have successfully fabricated concentric segmented 2D structures from molecules **1** and **2** (Fig. 1b). Specifically, thin hexagonal debris was first prepared from molecule **1** through a multi-step process, which involved the sonication of the original 2D platelets formed initially from molecule **1** (Supplementary Fig. 7a) the centrifugation for obtaining the thin sonicated debris (Supplementary Fig. 7b), and the subsequent seeded growth and further sonication (see the Method section for more details). The debris in acetonitrile (2.5 μL, 0.02 mg/mL) acquired through the above process was drop-cast onto a quartz slice, followed by the addition of 100 μL of **1** (0.02 mg/mL) in a chloroform/acetonitrile mixture (v/v: 1:5) to enable the seeded growth of thin hexagonal platelets. Notably, this particular solvent environment presents an appropriate level of supersaturation for both molecules **1** and **2**, thus restraining any spontaneous nucleation and permitting the growth of pre-existing seeds. Supplementary Fig. 8 reveals that the produced thin hexagonal platelets on a quartz slide have a uniform thickness of approximately 95 nm. These platelets were then employed as seeds to grow concentric segmented 2D structures, for which 100 μL of **1** and **2** (0.01 mg/mL) in a chloroform/acetonitrile mixture (v/v: 1:5) were added in succession to achieve the seeded growth over a certain period (Fig. 1b and Supplementary Fig. 9). The concentric multi-hexagonal 2D platelets with distinct alternative emissions were observed under fluorescence microscopy (Fig. 1c and Supplementary Fig. 10). The orange and yellow emissions were attributed to molecules **1** and **2**, respectively. The heteroepitaxial growth is likely due to the slight dissimilarity in the molecular structures of molecules **1** and **2**, which allows them to adopt similar molecular packing. Selected area electron diffraction characterizations confirmed the pretty similar molecular packing within the two distinct segments (Supplementary Fig. 11). The powder X-ray diffraction (XRD) analyses verified the consistent molecular packing (Supplementary Fig. 12). These XRD patterns aligned well with the single crystal data (Supplementary Fig. 12), demonstrating that the electrostatic attraction, in conjunction with π-interactions, plays a pivotal role in the development of a hexagonal 2D structure from molecules **1** and **2**[30]. Despite the formation of two distinct

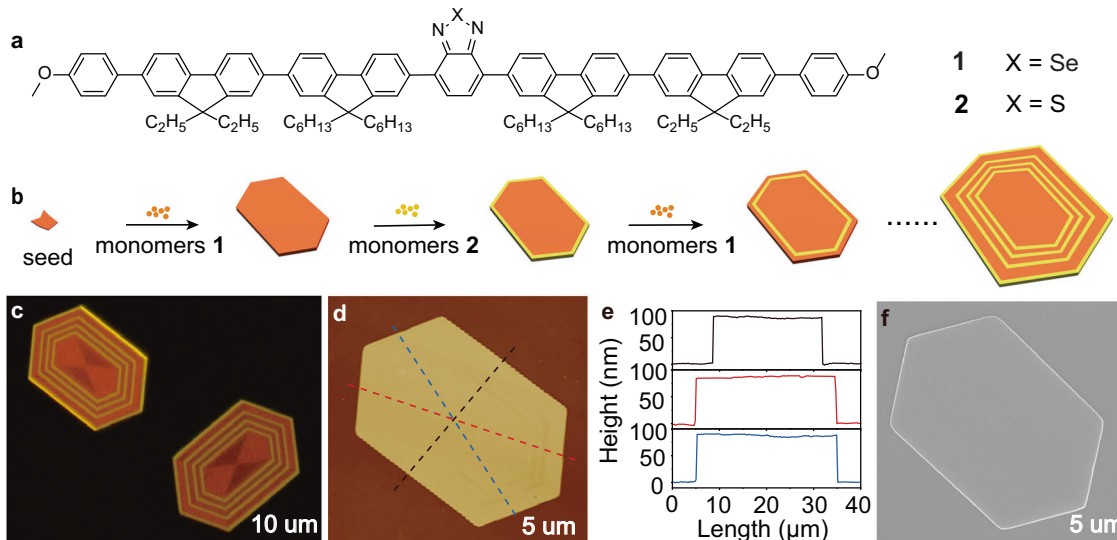

**Fig. 1 | Molecular structures and preparation of multi-hexagonal platelets.** **a** Molecular structures of molecules **1** and **2**. **b** Schematic diagram illustrating the seeded growth of concentric multi-hexagonal 2D platelets by the sequential, alternate addition of **1** and **2** monomers. **c** Fluorescence-mode optical microscopic images of two typical concentric multi-hexagonal 2D platelets (see detailed preparation in the Method section). **d** AFM height image and (**e**) the corresponding height profiles of a typical concentric multi-hexagonal 2D platelet. **f** SEM image of a typical concentric multi-hexagonal 2D platelet.

segments from molecules **1** and **2**, the multi-block 2D platelet exhibits a uniform flat surface, as confirmed by atomic force microscopy (AFM) (Fig. 1d, e) and scanning electron microscopy (SEM) imaging (Fig. 1f).

## Construction of hollow platelets via post-processing

Upon observation under fluorescence microscopy with a UV light source of 365 nm and 300 mW/cm$^2$, we noticed that the orange emission of the concentric multi-hexagonal 2D platelets gradually decreased (Supplementary Fig. 13). This implies that the segments of **1** are subject to photooxidation under ambient condition. The intensity of the platelets of molecular **2** as a function of UV irradiation time further confirms the occurrence of a photooxidation reaction (Supplementary Fig. 14). In comparison, the intensity of the segments of **2** only slightly changed under the same conditions (Supplementary Fig. 14).

To gain insight into the photooxidation of **1**, we conducted additional characterizations. Firstly, we employed matrix-assisted laser desorption/ionization time of flight mass spectrometry (MALDI-TOF-MS), a commonly used method for product identification, to analyze the products generated from UV irradiation of 2D platelets from **1**. As depicted in Supplementary Fig. 15-16, a series of oxidized products were identified, with the one containing four fluorenone moieties appearing to be the ultimate product. The results are consistent with the infrared (IR) analysis of the 2D platelets pre- and post-10-minute UV irradiation (365 nm, 300 mW/cm$^2$). Supplementary Fig. 17 illustrates the emergence of a band at 1720 cm$^{-1}$, attributed to fluorenone[31–33], in the IR spectrum obtained from the 2D platelets following UV exposure. Secondly, the photooxidation of 2D platelets assembled from **1** is conclusively linked to singlet oxygen. Specifically, singlet oxygen was significantly generated in the solution of 2D platelets assembled from **1** under UV irradiation, as validated by electron paramagnetic resonance (EPR) spectroscopy (Supplementary Fig. 18), aligning with their swift photooxidation process (Supplementary Fig. 14). In contrast, 2D platelets assembled from **2** under the same conditions exhibited a significantly lower level of singlet oxygen generation (Supplementary Fig. 18), consistent with their notably slower photooxidation rates (Supplementary Fig. 14). The results presented here are in agreement with prior observations that demonstrate the conversion of fluorene with two alkyl groups to fluorenone through singlet oxygen photooxidation[34]. The correlation between photooxidation and singlet oxygen was reinforced by the observation of a minimal presence of singlet oxygen in a toluene solution of molecules **1** and **2** (Supplementary Fig. 19), consistent with their proven high photostability, as illustrated in Supplementary Fig. 14. It is noteworthy that the spin trap (i.e., 5,5-Dimethyl-1-pyrroline N-Oxide) did not detect any superoxide radical, thereby eliminating the possibility of an oxidation mechanism involving this radical. The significant production of singlet oxygen in the presence of 2D structures, as opposed to the minimal generation of singlet oxygen in solution, emphasizes the crucial role of interactions among the benzoselenadiazole groups in molecule **1**. The intermolecular interactions in 2D structures, demonstrated by the red-shifted charge-transfer (CT) band in comparison to its isolated form (see Supplementary Fig. 20), may promote intersystem crossing (ISC) to the triplet state, possibly by enhancing the heavy atom effect. This, in turn, triggers the generation of singlet oxygen from molecular oxygen.

## Construction of concentric multi-hexagonal 2D hollow structures

It is anticipated that the increased polarity of the oxidized product of **1** will facilitate its solubility in polar solvents. This thus allows for the selective removal of the oxidized segments of **1** from the concentric multi-hexagonal 2D platelets, forming hollow multi-hexagonal 2D platelets. To verify this, we immersed the concentric multi-hexagonal 2D platelets into a chloroform/acetonitrile (v/v, 1/5) mixture and

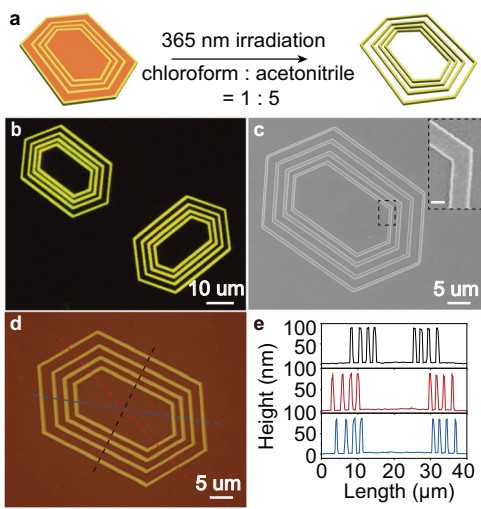

**Fig. 2 | Construction of concentric multi-hexagonal 2D hollow structures via post-processing. a** Schematic diagram illustrating the preparation of a hollow multi-hexagonal platelet which were obtained by immersing the concentric multi-hexagonal 2D precursors platelets in a chloroform/acetonitrile (v/v, 1/5) mixture (2 mL) and exposing to 365 nm UV irradiation (300 mW/cm$^2$) for 4 min to wash away the segments of **1**. **b** Fluorescence-mode optical microscopic image of two hollow multi-hexagonal platelets. **c** SEM image of a hollow multi-hexagonal platelet. Insert: zoomed-in image showing the smooth edge of hollow segments. Scale bar: 500 nm. **d** AFM height image and (**e**) corresponding height profiles of the hollow multi-hexagonal platelet shown in (**b**).

exposed them to UV irradiation for 4 min to dissolve molecule **1** slowly (Fig. 2a). Figure 2b demonstrated the successful production of hollow multi-hexagonal 2D platelets, with the orange-emissive regions of **1** being completely removed, leaving only the yellow-emissive regions of **2**. The hollow structure with smooth edges was confirmed by SEM imaging as shown in Fig. 2c. AFM measurements over the hollow platelets (Fig. 2d, e) further confirmed that the thickness of the remaining regions of **2** was the same as the pre-washed platelets, indicating that the washing process has no effect on the segment of **2**. Altogether, the above results demonstrate that the concentric multi-hexagonal 2D platelets can effectively serve as precursors for the preparation of hollow 2D structures through post-processing involving light irradiation and dissolution.

We next exploited the capacity of the methodology for complex concentric multi-hexagonal 2D hollow structures with increasing segment widths and numbers. As depicted in Fig. 3a and Supplementary Fig. 21, the multi-hexagonal 2D platelets with the increasing segment widths of **2** were achieved by gradually increasing the growth time of molecule **2** on the seeds. The corresponding hollow 2D structures were then prepared using a chloroform/acetonitrile (v/v, 1/5) mixture to wash away the protoxidized segments of **1** (Fig. 3b). AFM measurements (Fig. 3c, d) were conducted to further characterize the well-defined hollow structures, which revealed a uniform thickness of approximately 95 nm for all the segments. To further validate the robustness and controllability of the fabrication technique, we endeavored to create concentric hollow structures with an increased number of segments. As shown in Fig. 3e and Supplementary Fig. 22, concentric 2D platelets containing six segments of each component were successfully fabricated by extending the alternating cycles of the seeded growth of **1** and **2**. These precursors then enabled the preparation of hollow 2D structures with six hexagonal rings of **1** (Fig. 3f) using the above washing process. Likewise, the well-defined hollow structure was clearly confirmed by AFM measurements (Fig. 3g, h), which showed all six segments had a uniform thickness of around 95 nm. These results indicate the seeded heteroepitaxial growth

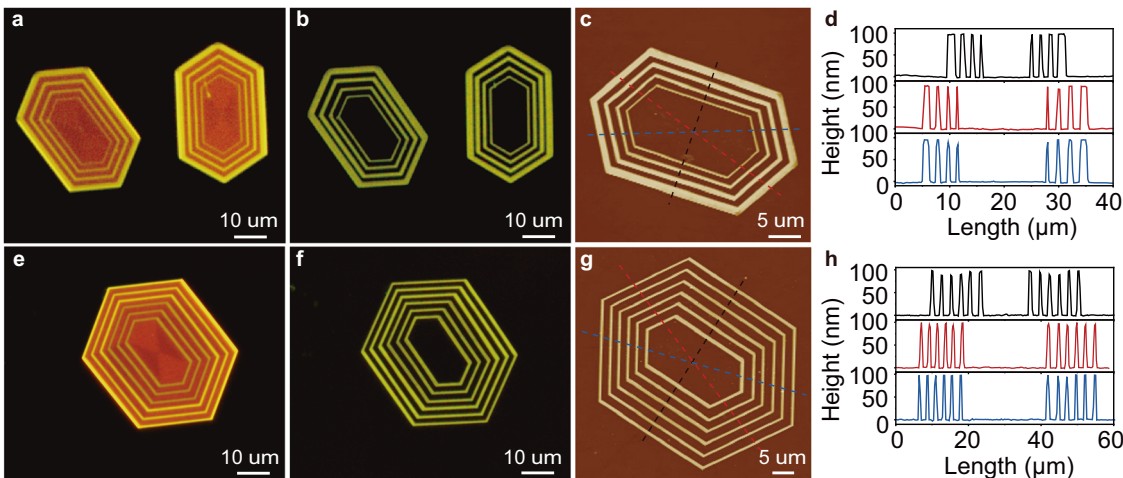

**Fig. 3 | Construction of complex concentric multi-hexagonal 2D hollow structures with increasing segment widths and numbers.** Fluorescence-mode optical microscopic images of (**a**) two concentric multi-hexagonal 2D platelets (see detailed preparation in the Method section) and (**b**) the corresponding hollow structures with increasing widths of segments of **2** which were obtained by immersing the concentric multi-hexagonal 2D precursors platelets in a chloroform/acetonitrile (v/v, 1/5) mixture (2 mL) and exposing to 365 nm UV irradiation (300 mW/cm²) for 4 min to wash away the segments of **1**. **c** AFM topography image and (**d**) the corresponding height profiles of the hollow platelet shown in (**b**). Fluorescence-mode optical microscopic images of (**e**) a concentric 2D platelets with six segments for each component (see detailed preparation in the Method section) and (**f**) the corresponding hollow structure with six segments which were obtained by immersing the concentric multi-hexagonal 2D precursors platelets in a chloroform/acetonitrile (v/v, 1/5) mixture (2 mL) and exposing to 365 nm UV irradiation (300 mW/cm²) for 4 min to wash away the segments of **1**. **g** AFM height image and (**h**) corresponding height profiles of the hollow platelet shown in (**f**).

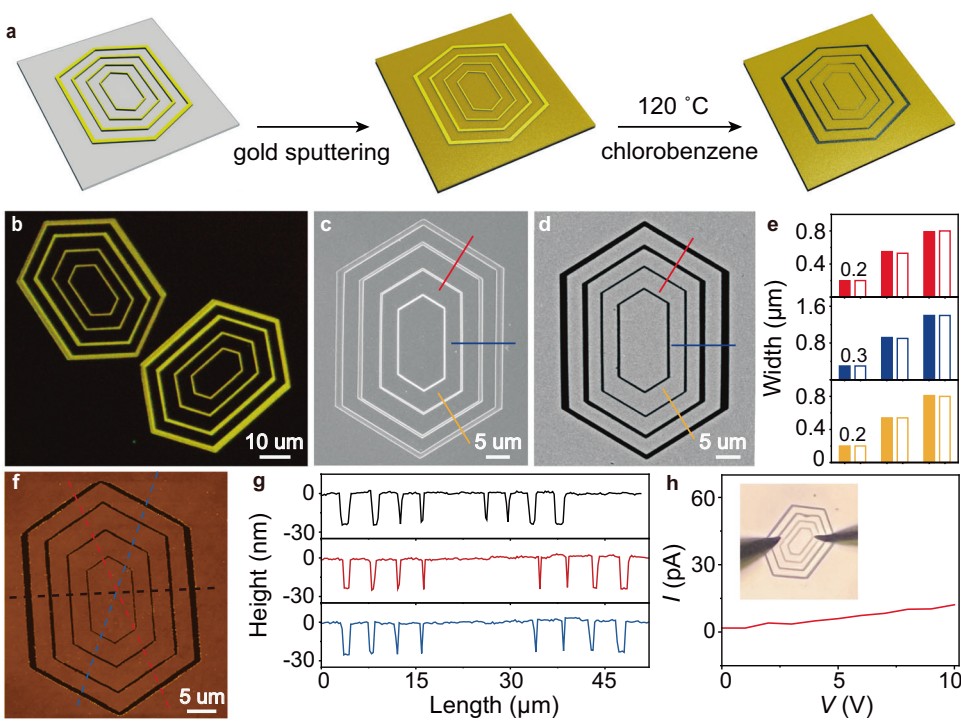

**Fig. 4 | Fabrication of intricate electrodes by using hollow platelets as templates. a** Schematic diagram illustrating the preparation of concentric electrodes using the hollow multi-hexagonal 2D platelets as the template. **b** Fluorescence-mode optical microscopic image of a hollow 2D platelets with varying segment widths which were obtained by immersing the concentric multi-hexagonal 2D precursors platelets in a chloroform/acetonitrile (v/v, 1/5) mixture (2 mL) and exposing to 365 nm UV irradiation (300 mW/cm²) for 4 min to wash away the segments of **1**. **c** SEM image of the hollow template on a quartz substrate after sputtering gold. **d** SEM image of the intricate electrode obtained after washing away the template. **e** Comparison between the segment widths of the template (solid) and the electrode gap (blank) fabricated therefrom. **f** AFM height image and (**g**) corresponding height profiles of the concentric electrodes shown in (**d**). **h** *I-V* curve measured over the electrodes with the smallest gap as shown in the insert.

method for the 2D precursors, combined with the post-processing approach involving selective photooxidation and dissolution, allows for precise control over both the widths and number of segments in the concentric hollow structures.

## Fabrication of intricate electrodes by using hollow platelets as templates

After successfully creating the hollow multi-hexagonal 2D platelets with varying widths and numbers of components, we studied the possibility of using them as templates for constructing intricate electrodes (Fig. 4a). By following the prescribed technique above, hollow 2D platelets composed of multi-hexagonal segments of molecule **2** was created, featuring a constant gap of about 3 μm and different segment widths (Fig. 4b and Supplementary Fig. 23). The platelets were then utilized as templates for intricate electrodes. After sputtering gold onto the templates on a quartz substrate (Fig. 4c), the templates were immersed in chlorobenzene at 120 °C for 24 hours, followed by a rinse with dichloromethane. SEM image confirmed that the template was completely washed away, thus producing a smooth edge (Fig. 4d and Supplementary Fig. 24). Notably, the concentric electrodes exhibit gap widths that align seamlessly with the segment widths in the original template along different directions (Fig. 4e), with the smallest gap of around 200 nm. The well-defined geometry and uniformness of the concentric electrodes were further characterized by AFM measurements (Fig. 4f, g), which indicate a uniform thickness of 25 nm for all the electrodes. The current-voltage (*I-V*) measurement of the electrodes over the smallest gap exhibited a typical Ohmic contact (Fig. 4h). On the basis of the linear *I-V* relationship, we estimated the electrical resistance across the gap to be ca. 670 GΩ. This value is consistent with the high insulating characteristics of quartz, confirming the complete removal of the template. These results validate the feasibility of utilizing the hollow 2D platelets as template to imprint concentric electrodes with variable gap widths. The concentric electrodes thus fabricated hold potential applications in investigating the optical and electrical properties of functional materials.

In conclusion, we have successfully prepared concentric segmented 2D platelets from two D-A molecules featuring adjustable segment widths and numbers using seeded heteroepitaxial growth techniques. The segments of molecule **1** containing the benzoselenadiazole group can produce a much higher level of singlet oxygen in comparison with UV irradiation of the segment of **2**, under UV irradiation, so that the segments can be easily photooxidized and then washed away by polar solvents to form concentric hollow 2D nanostructures. It is worth noting that the combination of seed heteroepitaxial growth and selective photooxidation can produce hollow multi-hexagonal 2D sheets with programmable widths and number of segments. These well-defined hollow 2D platelets can serve as templates for the fabrication of complex plane electrodes with precisely controlled gap widths, with a minimum gap of about 200 nm, which would provide a convenient platform for exploring the optoelectronic properties of functional materials.

## Methods

### Synthetic chemistry

The synthetic routes of the molecules **1** and **2** are shown in Supplementary Figs. 1, 4. The detailed characterization of molecules **1** and **2** are described in Supplementary Figs. 2–3, 5 and 6.

### Preparation of mono-component thin hexagonal platelets via living seeded self-assembly

Thin hexagonal platelets were prepared from molecule **1** through a multi-step process. Initially, the original hexagonal platelets underwent sonication, followed by centrifugation to obtain the thin sonicated debris. The debris obtained was then used as seed to grow into larger thin platelet.

The original hexagonal platelets of **1** were prepared by injecting 1 mL of acetonitrile into 0.2 mL of a chloroform solution of **1** (0.5 mg/mL) in a 4 mL vial, followed by aging at 25 °C for 40 h. The resulting hexagonal platelets suspended in 2 mL of acetonitrile were sonicated at −35 °C for 5 min, followed by centrifugation at 2270 x g for 10 min. The irregular debris in the upper layer with unspecified quantity were then collected and used as seeds for growing thin hexagonal platelets. To obtain a defined quantity of thin seeds, a minute volume of the resulting thin debris (20 μL) was next added to 1.2 mL of **1** (0.02 mg/mL) in a chloroform/acetonitrile mixture (v/v: 1:5) in a 4 mL vial and allowed seeded growth at 25 °C for 5 min, which yielded quantitatively thin hexagonal platelets (0.024 mg). Subsequently, these hexagonal platelets (0.024 mg) in the same solution were then sonicated through the method described above to quantitatively produce thin debris as seeds (0.02 mg/mL, 1.2 mL). The thin debris dispersed in acetonitrile (2.5 μL, 0.02 mg/mL) were drop-cast onto a quartz slip, followed by adding 100 μL of **1** (0.02 mg/mL) in a chloroform/acetonitrile mixture (v/v: 1:5) atop the seed layer to grow the thin hexagonal platelets as the cores of the concentric multi-segment 2D platelets.

### Preparation of concentric multi-hexagonal 2D platelets with controllable size via living seeded self-assembly

The concentric multi-hexagonal 2D platelets were prepared by alternatively adding 100 μL of **1** and **2** (0.01 mg/mL) into a chloroform/acetonitrile mixture (v/v: 1:5) drop cast on a quartz slip containing the pre-fabricated thin hexagonal platelets of **1** as the seeds. By controlling the growth time of molecules **1** and **2** on these seeds, the concentric multi-hexagonal 2D platelets with different segment widths were obtained. After each growing cycle, the remaining solution were washed away with acetonitrile before the new solution of **1** or **2** was added. Upon setting the alternative growth time as 20 s, 30 s, 30 s, 50 s, 35 s, 80 s, and 40 s, the concentric multi-hexagonal 2D platelets as shown in Fig. 1c were prepared. Likewise, by setting the alternative growth times time as 15 s, 45 s, 25 s, 70 s, 50 s, 75 s, and 95 s, we obtained the concentric multi-hexagonal 2D platelets shown in Fig. 3a; by setting the alternative growth times time as 45 s, 60 s, 54 s, 85 s, 75 s, 90 s, 80 s, 105 s, 96 s, 100 s, and 115 s, we obtained the concentric multi-hexagonal 2D platelets shown in Fig. 3e; by setting the alternative growth times time as 30 s, 150 s, 45 s, 250 s, 120 s, 350 s, and 180 s, we obtained the concentric multi-hexagonal 2D platelets shown in Supplementary Fig. 23.

### Fabrication of the concentric multi-hexagonal electrodes by using the hollow platelets as templates

By following the procedure prescribed above, the hollow 2D platelets composed of multi-hexagonal segments of molecule **2** were fabricated, featuring a constant gap of about 3 μm and varying segment widths. The hollow platelets were then utilized as templates for preparing intricate electrodes. Briefly, a gold film was sputter-coated onto the templates pre-fabricated on a quartz substrate. The coated sample was then immersed in chlorobenzene at 120 °C for 24 h to remove the template, followed by rinsing with dichloromethane.

### Property characterizations

Fluorescence microscopic images were obtained using an inverted fluorescence microscope (Olympus X71). Scanning electron microscopy (SEM) images were recorded on a Hitachi S-8010. Atomic force microscopy (AFM) images were obtained using a Nanowizard NanoOptics atomic force microscope. Selected area electron diffraction (SAED) was acquired using JEOL 2100 with an electron beam energy of 120 kV. Powder XRD measurements were performed on a PAN alytical X'Pert PRO instrument (40 kV, 200 mA). The photostability was evaluated by monitoring the fluorescence intensity of toluene solution of the two molecules and platelets from the two

molecules as a function of UV irradiation time, which was obtained using an Ocean Optics USB4000 fluorometer with a 385 nm LED lamp (3 mW/cm$^2$) as the excitation light source. UV-vis absorption spectra of the solution (0.02 mg/mL) and platelets were obtained on a Hitachi U-3900 spectrometer. The fluorescent spectra were collected using a lambda-mode option of the Olympus FV1000 inverted confocal laser scanning fluorescence microscopy, where the samples were excited by a continuous wave laser (405 nm). The coating of templates with thin and fine-grained gold films was carried out with a high vacuum sputter coater (LEICA EM SCD 500). The current-voltage (*I-V*) measurements of the concentric electrodes were carried out through a two-probe method with a Signatone S-1160 probe station coupled with an Agilent B1500A Precision semiconductor parameter analyzer. The probe station was equipped with a Motic microscope for in-situ imaging and positioning. The sonication of platelets to prepare small seeds (debris) was conducted using a commercial SCIENTZ-IID ultrasonic homogenizer at a power level of 100 W. EPR spectra of singlet oxygen generated upon exposure the 2D platelets suspended in an acetonitrile solution (0.5 mg/mL) or a toluene solution of molecules **1** and **2** (0.5 mg/mL) to UV irradiation (365 nm, 300 mW/cm$^2$) were obtained on a Bruker EPR ELEXSYS E500 spectrometer using triacetonamine hydrochloride (10 mM) and 5,5-Dimethyl-1-pyrroline N-Oxide (5 mM) as the spin trap of singlet oxygen and superoxide radical, respectively. Microwave power was set s at 13 dB, and the sweep time was 81.92 ms.

## Data availability

The raw data for each curve in Figs. 1e, 2e, 3d, 3h, 4e, 4g, 4h and Supplementary Figs. generated in this study are provided in the Source Data file. Crystallographic data for the structures reported in this Article have been deposited at the Cambridge Crystallographic Data Centre, under deposition numbers CCDC 2261026. Copies of the data can be obtained free of charge via https://www.ccdc.cam.ac.uk/structures/.' Source data are provided with this paper.

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

## Acknowledgements

We thank. National Key Research and Development Program of China (No. 2022YFA1205501, 2019YFA0210401), NSFC (Nos. 22321004, 21925604) and the Strategic Priority Research Program of Chinese Academy of Sciences (Grant No. XDB36000000).

## Author contributions

Yanke C. conceived the project. C.L. and Yanxue C. synthesized the materials. C.L. and Y.G. performed the experiments with assistance from other authors. C.L., Y.G., L.Z. and Yanke C. analyzed the data. B.L., L.Z., and Yanke C. wrote the manuscript. C.L., Y.G., Yanxue C., H.J., B.L., L.Z., Yanke C. and J.Z. discussed the results and commented on the manuscript.

## Competing interests

The authors declare no competing interests. Yanxue Che is employed at HT-NOVA Co., Ltd and declares no competing interest.
