## [Peer Review File · Nature Communications]

Concentric Hollow Multi-Hexagonal Platelets from A Small MoleculeEditorial Note: Parts of this Peer Review File have been redacted as indicated to remove third-party material where no permission to publish could be obtained.

REVIEWER COMMENTS

Reviewer #1 (Remarks to the Author):

The authors report the fabrication of well-defined 2D segmented platelets using seeded heteroepitaxial growth strategy by using two chemically-distinct small molecules. Upon UV irradiation, one of the molecules can be selectively oxidized, and then this platelet domain could be selectively removed by solvent dissolution, thus 2D hollow platelets would be simply created.

The authors present an interesting study on formulating unique 2D hollow platelets by light irradiation. Their experimental design and analysis are thorough in determining the ability for the 2D hollow platelets. The use of microscopy techniques provides ample evidence of the mechanism for selective removal of the specific regions. The manuscript is well-presented and the reviewer has only some minor comments to the manuscript.

Similar small molecules have been synthesized by the group previously and well-defined segmented platelets have been fabricated via seeded heteroepitaxial growth (J. Am. Chem. Soc., 2022, 144, 33, 15403 (ref. 9); J. Am. Chem. Soc., 2023, 145, 17, 9771 (not cited)). Hence, it is important to include a statement why small molecules 1 and 2 are selected and the advance of present work compared to previous works.

Concerning the review of relevant works in the Introduction, works on 2D segmented micelles of PCL core-forming block should be added, as the corona chemistry or core-topology can be tailored in terms of functionality, and also hierarchical superstructures are accessible (Adv. Mat. 2023, 2308154; J. Am. Chem. Soc. 2023, 145, 28049; Macromolecules 2023, 56, 9685; Macromolecules 2022, 55, 1067; Macromolecules 2022, 55, 8250)

Other minor comments:

1. The original platelets of 1 and the sonicated seeds should be also added in the SI.
2. The XRD profiles of 1 and 2 should be added to confirm the crystalline structure of two molecules meets the requirement of lattice matching.
3. More information should be added for the analysis of SAED, i.e., the d-spacing values determined from two regions.
4. It seems that the 2D elongated hexagonal platelets of 1 show a unique fluorescence surface patterning (Figure 1c, most inner platelet domain), i.e., the (110) and (200) platelet domains show different fluorescence colors. Is this interesting result due to the different molecular arrangements in those two crystalline domains? Can the authors add some information for this?
5. Why the remaining platelet domain 2 would not slide out from its place after platelet domain of 1 are completely removed by solvent solution. Can the authors add some information for this?
6. For the cited references: "The first step can be fairly addressed by seeded heteroepitaxy growth techniques that has been used for the construction of complex multi-component solid 2D nanomaterials 7-21" The authors describe the seeded heteroepitaxial growth techniques here, but

the references from 7-21 are not all related to the heteroepitaxial growth and many of them are homoepitaxial growth from one identical crystallizable polymer. The author should be cited relevant references here and remove the homoepitaxial growth references to other places. Moreover, some important works about seeded heteroepitaxial growth of block copolymers are missing and they are recommended to be added (Macromolecules 2023, 56, 5984; ACS Nano 2023, 17, 24141).

Some other comments:

1. In the abstract, the first appearance of 2D should be defined.
2. In the first paragraph, copolymes should be “copolymers”

Reviewer #2 (Remarks to the Author):

The paper reports the synthesis of hollow nanostructure by synthesizing the multiblock comicelles followed by selective dissolution of the middle blocks. This is enabled by the selective photooxidation of the benzoselenadiazole-containing small molecule, which can be done easily by irradiating 365 nm UV light for 4 min. The authors have also shown that this hollow structures can be further used as the template for intricate electrode fabrication, by gold sputtering and dissolving out the template.

It seems like a very clever design to easily prepare the hollow structures, and also is very interesting that this demonstrates the possibility of further application. However, I think there are some major and minor things to be further elucidated before accepting it for the publication.

- The characterization of photooxidation and the reasoning for this reaction seems weak. The authors claim that fluorenone is formed only for 1, but the two experimental data are suggested for the support; one is the appearance of the C=O stretching peak in IR spectrum (which does not necessarily mean that fluorenone is formed), and the other is the disappearance of the fluorescence signal only for 1, which also does not directly support the fluorenone formation. Also, as far as I know, fluorenone is fluorescent as well. Running the photooxidation reaction just by irradiating light to the small molecule solution or maybe solid, and characterizing the product by NMR will give a much better clue on what is actually happening under UV irradiation. Just based on the current manuscript, I cannot accept the argument on relating the Se–N and S–N bond length and the oxidation reaction of 9-alkylated fluorene to form fluorenone.

- From figure 3, the emission color before and after dissolving out the 1 layer looks somewhat different, although the emission spectra in the SI before and after UV irradiation is identical for 2. I wonder if there really is a color difference, and if there is, what makes the difference.

- The nanostructure formation step is longer than the conventional seeded growth approach. Sonication is applied twice, and I wonder the reason.

- There is a typo in the main text; AFM is written as “aromatic” force microscopy.

- There are few more relevant papers on 2D nanosheets from conjugated polymers which could be cited, Nat. Commun. 9, 865 (2018); J. Am. Chem. Soc. 139, 3082–3088 (2017); J. Am. Chem. Soc. 141, 19138–19143 (2019).

Reviewer #3 (Remarks to the Author):

In this work, Liao and coworkers demonstrated a photo-induced method for preparing concentric hollow multi-hexagonal 2D platelets with controlled widths from two donor-acceptor (D-A) molecules (**1** and **2**), which contain fluorene and benzimidazole groups as the D units and benzothiadiazole or benzoselenadiazole groups as the A units. First, solid multi-hexagonal 2D platelets of **1** and **2** were generated following a well-established alternate heteroepitaxial seeding technique. Subsequently, UV irradiation was used to selectively oxidize one of the two D-A molecules, i.e., benzoselenadiazole-based compound **1**, making the resulting product more soluble due to enhanced polarity. Solvent dissolution could selectively remove oxidized segments, resulting in hollow multiblock 2D structures. Further, the authors have used those hollow concentric rings as templates for constructing intricate electrodes by sputtering gold onto these templates, followed by immersing in chlorobenzene at 120 °C and rinsing with dichloromethane. Similar small molecules for the construction of different types of 2D platelets, including hexagonal morphology, by seeding techniques have been previously reported by this group (*Chem. Eur. J.* 2023, 29, e202301747; *J. Am. Chem. Soc.* 2022, 144, 15403–15410; *J. Am. Chem. Soc.* 2023, 145, 9771–9776). The concept of well-controlled multi-component 2D platelets by seeding and their selective dissolution to form hollow structures is quite established (*J. Am. Chem. Soc.* 2017, 139, 9221–9228; *Nat. Chem.* 2023, 15, 824–831; *Science*, 2016, 352, 697–701). Overall, the work is interesting, but unfortunately, the study on hollow 2D platelets as templates to imprint concentric electrodes is premature and not explored in detail in this paper. In my opinion, this work lacks novelty and impact for the journal *Nat. Commun.* Here are some specific comments:

Unfortunately, this part has not been explored in this report.

- The oxidation mechanism of fluorene to fluorenone needs clarity. The 9th position (active site) of fluorene is occupied with two alkyl groups, so it is not so easy to do oxidation in that position, and the reason for oxidation described in the paper is not convincing enough. The benzoselenadiazole core is far apart from the terminal fluorene groups. How does a slight difference in the Se-N bond impact the selective photooxidation of compound **1** in a highly packed state?
- A detailed study of the chemistry happening upon UV irradiation must be conducted with other control molecules. The NMR technique may be used to demonstrate the formation of the oxidized species. I wonder what happens when the UV irradiation time increases.
- Selected area electron diffraction characterizations have not shed any light on the specification of unit cell parameters. The difference in the SAED pattern may be compared before and after the hollow structure formation.
- There is no clear explanation of what the driving force is for forming such a precise, controlled hexagonal 2D structure on a quartz slice in a few seconds.

- Why is there a discrepancy between the platelet thickness (95 nm) and the thickness (25 nm) of all the electrodes? The extent of sputtering will probably define the thickness of the gold deposition, which is independent of the uniform platelet thickness. From the platelet thickness of 95 nm, it seems like they are not monolayered. How is it possible to control the pi-stacking distance to obtain the uniform thickness of the 2D surface?

Point-by-point response to the reviewers' comments

Reviewer #1 (Remarks to the Author):

The authors report the fabrication of well-defined 2D segmented platelets using seeded heteroepitaxial growth strategy by using two chemically-distinct small molecules. Upon UV irradiation, one of the molecules can be selectively oxidized, and then this platelet domain could selectively removed by solvent dissolution, thus 2D hollow platelets would be simply created. The authors present an interesting study on formulating unique 2D hollow platelets by light irradiation. Their experimental design and analysis are thorough in determining the ability for the 2D hollow platelets. The use of microscopy techniques provides ample evidence of the mechanism for selective remove of the specific regions. The manuscript is well-presented and the reviewer has only some minor comments to the manuscript.

Response: We are thankful to the reviewer for their positive feedback in general.

Similar small molecules have been synthesized by the group previously and well-defined segmented platelets have been fabricated via seeded heteroepitaxial growth (J. Am. Chem. Soc., 2022, 144, 33, 15403 (ref. 9); J. Am. Chem. Soc., 2023, 145, 17, 9771 (not cited)). Hence, it is important to include a statement why small molecules 1 and 2 are selected and the advance of present work compared to previous works.

Response: Following the comment, we have provided the justification for the selection of molecules **1** and **2** in the revised MS. Additionally, we have expounded on the progress made in our study relative to previous studies. In brief, the D'DADD' molecules, as reported in a recent study (J. Am. Chem. Soc., 2022, 144, 33, 15403), feature the benzimidazole moieties as the end group D', which engages in hydrogen bonding with alcohol to facilitate the formation of 2D structures. The persistent hydrogen bonding with alcohol within these 2D structures poses a barrier to the photooxidation of molecules containing benzothiadiazole and benzoselenadiazole groups. Consequently, these 2D structures are unsuitable for the generation of hollow structures. In order to address this matter, we utilized molecules **1** and **2**, which lack benzimidazole moieties but include fluorene groups as the D part. In contrast to the benzothiadiazole group in **2**, the benzoselenadiazole group in **1**, is anticipated to facilitate intersystem crossing (ISC) for triplet state formation. Consequently, this process is expected to produce singlet oxygen, leading to the oxidation of fluorene units in **1** and thereby imparting unique characteristics for subsequent post-processing toward hollow structures.

Concerning the review of relevant works in the Introduction, works on 2D segmented micelles of PCL core-forming block should be added, as the corona chemistry or core-topology can be tailored in terms of functionality, and also hierarchical superstructures are accessible (Adv. Mat. 2023, 2308154; J. Am. Chem. Soc. 2023, 145, 28049; Macromolecules 2023, 56, 9685; Macromolecules 2022, 55, 1067; Macromolecules 2022, 55, 8250)

Response: We are thankful to the Reviewer for the comment. We have added these refs in the revised MS.

Other minor comments:

1. The original platelets of 1 and the sonicated seeds should be also added in the SI.

Response: Following the suggestion, we have added the fluorescence-mode optical microscopic images of the original platelets of **1** and the SEM image of the sonicated seeds as **Figure S3** in the revised supporting information.

2. The XRD profiles of 1 and 2 should be added to confirm the crystalline structure of two molecules meets the requirement of lattice matching.

Response: Following the recommendation, we have added the XRD profiles of **1** and **2** as **Figure S8** in the revised supporting information, which confirm the matching crystalline structure of two molecules. The related description has also been added in the revised MS.

3. More information should be added for the analysis of SAED, i.e., the d-spacing values determined from two regions.

Response: As suggested, we have added the related d-spacing values in the SAED figures (**Figure S7**).

4. It seems that the 2D elongated hexagonal platelets of 1 show a unique fluorescence surface patterning (Figure 1c, most inner platelet domain), i.e., the (110) and (200) platelet domains show different fluorescence colors. Is this interesting result due to the different molecular arrangements in those two crystalline domains? Can the authors add some information for this?

Response: We thank the reviewer for the comment. The variation in self-assembly rates across distinct domains is anticipated to generate heterogeneous packing defect in

various regions, consequently leading to variations in fluorescence between the (110) and (200) platelet domains. This analogous phenomenon has been previously documented in the literature (Adv. Mater. 2017, 29, 1605043). In the revised manuscript, we have included additional descriptions for this aspect.

5. Why the remaining platelet domain 2 would not slid out from its place after platelet domain of 1 are completely removed by solvent solution. Can the authors add some information for this?

Response: The concentric multi-hexagonal 2D platelets prepared by a drop-casting method can tightly adhere to the silica surface due to the relative strong binding. Upon UV irradiation, platelet domain 1 can be selectively oxidized and dissolved by a polar solvent, while platelet domain of 2 undergoes no reaction and therefore cannot be dissolved by a polar solvent, thereby sticking on a silica surface.

6. For the cited references: “The first step can be fairly addressed by seeded heteroepitaxy growth techniques that has been used for the construction of complex multi-component solid 2D nanomaterials 7-21” The authors describe the seeded heteroepitaxial growth techniques here, but the references from 7-21 are not all related to the heteroepitaxial growth and many of them are homoepitaxial growth from one identical crystallizable polymer. The author should be cited relevant references here and remove the homoepitaxial growth references to other places. Moreover, some important works about seeded heteroepitaxial growth of block copolymers are missing and they are recommended to be added (Macromolecules 2023, 56, 5984; ACS Nano 2023, 17, 24141).

Response: We are thankful to the reviewer for the comment. we have revised the references cited wrongly and have added the corresponding references in the revised manuscript.

Some other comments:

1. In the abstract, the first appearance of 2D should be defined.

Response: We thank the reviewer for pointing this out. We have revised it in the abstract.

2. In the first paragraph, copolymes should be “copolymers”

Response: We have corrected this typo in the revised manuscript.

Reviewer #2 (Remarks to the Author):

The paper reports the synthesis of hollow nanostructure by synthesizing the multiblock comicelles followed by selective dissolution of the middle blocks. This is enabled by the selective photooxidation of the benzoselenadiazole-containing small molecule, which can be done easily by irradiating 365 nm UV light for 4 min. The authors have also shown that this hollow structures can be further used as the template for intricate electrode fabrication, by gold sputtering and dissolving out the template.

It seems like a very clever design to easily prepare the hollow structures, and also is very interesting that this demonstrates the possibility of further application. However, I think there are some major and minor things to be further elucidated before accepting it for the publication.

Response: We acknowledge the reviewer's comments and have taken them into careful consideration to enhance our manuscript as detailed below.

*1. The characterization of photooxidation and the reasoning for this reaction seems weak. The authors claim that fluorenone is formed only for **1**, but the two experimental data are suggested for the support; one is the appearance of the C=O stretching peak in IR spectrum (which does not necessarily mean that fluorenone is formed), and the other is the disappearance of the fluorescence signal only for **1**, which also does not directly support the fluorenone formation. Also, as far as I know, fluorenone is fluorescent as well. Running the photooxidation reaction just by irradiating light to the small molecule solution or maybe solid, and characterizing the product by NMR will give a much better clue on what is actually happening under UV irradiation. Just based on the current manuscript, I cannot accept the argument on relating the Se–N and S–N bond length and the oxidation reaction of 9-alkylated fluorene to form fluorenone.*

Response: In agreement with the Reviewer, we acknowledge the need to strengthen the characterization of photooxidation and provide a clearer rationale for this reaction. In this regard, we have conducted additional characterizations to clarify the photooxidation mechanism. Firstly, we employed matrix-assisted laser desorption/ionization time of flight mass spectrometry (MALDI-TOF-MS), a commonly used method for product identification, to analyze the products generated from UV irradiation of 2D platelets from **1**. As depicted in Figure S11-12, a series of oxidized products were identified, with the one containing four fluorenone moieties

appearing to be the ultimate product. The results are consistent with the infrared (IR) analysis of the 2D platelets pre- and post-10-minute UV irradiation (365 nm, 300 mW/cm²). Figure S13 illustrates the emergence of a novel band at 1720 cm⁻¹, attributed to fluorenone, in the IR spectrum obtained from the 2D platelets following UV exposure. Secondly, the photooxidation of 2D platelets assembled from **1** is conclusively linked to singlet oxygen. Specifically, singlet oxygen was significantly generated in the solution of 2D platelets assembled from **1** under UV irradiation, as validated by electron paramagnetic resonance spectroscopy (Figure S14), aligning with their swift photooxidation process (Figure S10). In contrast, 2D platelets assembled from **2** under the same conditions exhibited a significantly lower level of singlet oxygen generation (Figure S14), consistent with their notably slower photooxidation rates (Figure S10). The results presented here are in agreement with prior observations that demonstrate the conversion of fluorene with two alkyl groups to fluorenone through singlet oxygen photooxidation (Adv. Mater. 2009, 21, 597-602). The correlation between photooxidation and singlet oxygen was reinforced by the observation of a minimal presence of singlet oxygen in a toluene solution of molecules **1** and **2** (Figure S15), consistent with their proven high photostability, as illustrated in Figure S10. The significant production of singlet oxygen in the presence of 2D structures, as opposed to the minimal generation of singlet oxygen in solution, emphasizes the crucial role of interactions among the benzoselenadiazole groups in molecule **1**. The intermolecular interactions in 2D structures, demonstrated by the red-shifted charge-transfer (CT) band in comparison to its isolated form (see Figure S16), may promote intersystem crossing (ISC) to the triplet state, possibly by enhancing the heavy atom effect. This, in turn, triggers the generation of singlet oxygen from molecular oxygen. We have added the new data in the revised supporting information as **Figure S10-12 and S14-16** and added more discussion and description in the revised MS to clarify the photooxidation mechanism.

*2. From figure 3, the emission color before and after dissolving out the **1** layer looks somewhat different, although the emission spectra in the SI before and after UV*

irradiation is identical for 2. I wonder if there really is a color difference, and if there is, what makes the difference.

Response: The emission spectra of the **2** layers are unchanged before and after dissolving out the **1** layer. The subtle variation in emission color shown in Figure 3 is attributed solely to vision disparity in the presence or absence of emission color in the **1** layer and slight fluctuations in exposure time during the acquisition of the fluorescent image.

3. The nanostructure formation step is longer than the conventional seeded growth approach. Sonication is applied twice, and I wonder the reason.

Response: We appreciate the reviewer's feedback. Sonication was employed twice to obtain a quantity of thin seeds. Initially, sonication and centrifugation were utilized to isolate thin debris in the upper layer, the amount of which was unspecified. To obtain a defined quantity of thin seeds, a minute volume of the resulting thin debris (20 μL) was combined with 1.2 mL of a solution of **1** (0.02 mg/mL) in a chloroform/acetonitrile mixture (v/v: 1:5) and allowed to undergo seeded growth at 25 °C for 5 minutes, which yielded quantitatively thin hexagonal platelets (0.024 mg). Subsequently, these hexagonal platelets (0.024 mg) were then sonicated through the method described above to quantitatively produce thin debris as seeds (0.02 mg/mL, 1.2 mL). We have revised the related description in the supporting information to enhance clarity.

4. There is a typo in the main text; AFM is written as “aromatic” force microscopy.

Response: We are grateful for the reviewer's reminder. In the revised manuscript, the typo has been corrected.

5. There are few more relevant papers on 2D nanosheets from conjugated polymers which could be cited, Nat. Commun. 9, 865 (2018); J. Am. Chem. Soc. 139, 3082–3088 (2017); J. Am. Chem. Soc. 141, 19138–19143 (2019).

Response: Following the suggestion, the references have been integrated into the revised manuscript.

Reviewer #3 (Remarks to the Author):

In this work, Liao and coworkers demonstrated a photo-induced method for preparing concentric hollow multi-hexagonal 2D platelets with controlled widths from two donor-

acceptor (D-A) molecules (1 and 2), which contain fluorene and benzimidazole groups as the D units and benzothiadiazole or benzoselenadiazole groups as the A units. First, solid multi-hexagonal 2D platelets of 1 and 2 were generated following a well-established alternate heteroepitaxial seeding technique. Subsequently, UV irradiation was used to selectively oxidize one of the two D-A molecules, i.e., benzoselenadiazole-based compound 1, making the resulting product more soluble due to enhanced polarity. Solvent dissolution could selectively remove oxidized segments, resulting in hollow multiblock 2D structures. Further, the authors have used those hollow concentric rings as templates for constructing intricate electrodes by sputtering gold onto these templates, followed by immersing in chlorobenzene at 120 °C and rinsing with dichloromethane. Similar small molecules for the construction of different types of 2D platelets, including hexagonal morphology, by seeding techniques have been previously reported by this group (*Chem. Eur. J.* 2023, 29, e202301747; *J. Am. Chem. Soc.* 2022, 144, 15403–15410; *J. Am. Chem. Soc.* 2023, 145, 9771–9776). The concept of well-controlled multi-component 2D platelets by seeding and their selective dissolution to form hollow structures is quite established (*J. Am. Chem. Soc.* 2017, 139, 9221–9228; *Nat. Chem.* 2023, 15, 824–831; *Science*, 2016, 352, 697–701). Overall, the work is interesting, but unfortunately, the study on hollow 2D platelets as templates to imprint concentric electrodes is premature and not explored in detail in this paper. In my opinion, this work lacks novelty and impact for the journal *Nat. Commun.* Here are some specific comments:

Response: We thank the reviewer for his/her comments. But several key points need clarification. First, the established concept of controlled multi-component 2D platelets through seeding and selective dissolution to create hollow structures is documented for polymer literature (*Science*, 2016, 352, 697–701; *J. Am. Chem. Soc.* 2017, 139, 9221–9228) rather than small molecules. The fabrication of these polymer-based hollow structures involves multiple synthetic steps, such as crosslinking the outer region and dissolving the central crosslink-free area (as noted in *Nat. Chem.* 2023, 15, 824–831). A recent innovative approach has emerged, which entails the selective degradation of polylactone-based polymers to eliminate time-consuming post-modification processes (*Nat. Chem.* 2023, 15, 824–831). Illustrated in the subsequent figure, each step following the initial concept, aimed at overcoming the technical challenge of accessing hollow 2D platelets from polymers, is original and has been published in a prestigious journal.

[REDACTED]

Second, when compared to polymers, small molecules encounter two primary challenges: the absence of an effective seeded heteroepitaxial growth method for producing well-defined size-controlled 2D structures (see the following figure), and the absence of techniques like polymers to selectively remove specific regions to create hollow structures. Over the past two years, our group has developed novel seeded methods for fabricating 2D structures using D-A molecules, as mentioned by the Reviewer (J. Am. Chem. Soc. 2022, 144, 15403–15410; J. Am. Chem. Soc. 2023, 145, 9771–9776; Chem. Eur. J. 2023, 29, e202301747). Specifically, a subset of these D-A molecules incorporates benzimidazole moieties as the terminal group D', which participate in continuous hydrogen bonding with a cofomer (alcohol) to promote 2D structure formation (J. Am. Chem. Soc., 2022, 144, 33, 15403; J. Am. Chem. Soc. 2023, 145, 9771–9776). The persistent hydrogen bonding with alcohol within these 2D structures hinders the selective removal of molecules containing benzothiadiazole and benzoselenadiazole groups. Consequently, the 2D structures produced through seeded heteroepitaxial growth are not suitable for generating hollow structures. A novel seeded growth method is required to produce well-defined 2D platelets without the use of a cofomer. We then developed a method for growing precise 2D platelets from a D-A molecule without cofomers (Chem. Eur. J. 2023, 29, e202301747). However, selectively removing specific regions in 2D heterojunctions to achieve hollow structures remains a significant challenge. Each step prior to this study, as depicted in the accompanying figure, is crucial and has been reported in reputable journals.

In this study, we synthesized 2D heterojunctions using two small D-A molecules, where one molecule has been found to be selectively photooxidized and subsequently removed through solvation. The creation of well-defined size-controlled hollow 2D structures from small molecules, for the first time, holds great potential for various applications, such as using 2D platelets as templates for fabricating concentric electrodes via photolithography. This research marks a significant advancement in the seeded self-assembly of small molecules. Notably, the fabrication of intricate electrodes underscores the potential of these hollow structures. A comprehensive investigation of this aspect, however, is beyond the scope of the current study.

Unfortunately, this part has not been explored in this report.

1. The oxidation mechanism of fluorene to fluorenone needs clarity. The 9th position (active site) of fluorene is occupied with two alkyl groups, so it is not so easy to do oxidation in that position, and the reason for oxidation described in the paper is not convincing enough. The benzoselenadiazole core is far apart from the terminal fluorene groups. How does a slight difference in the Se-N bond impact the selective photooxidation of compound 1 in a highly packed state?

Response: In agreement with the Reviewer, we acknowledge the need to strengthen the characterization of photooxidation and provide a clearer rationale for this reaction. In this regard, we have conducted additional characterizations to clarify the photooxidation mechanism. Firstly, we employed MALDI-TOF-MS, a commonly utilized method for identifying various products, to analyze the products resulting from UV irradiation of 2D platelets assembled from **1**. The corresponding data has been included as Figure S11-12 in the updated supplementary information. A series of step-by-step oxidized products has been identified, with the one containing four fluorenone moieties appearing to be the final product. It is noteworthy that, in contrast to the MALDI-TOF-MS results, the NMR spectrum of the multiple products presented convoluted data that proved challenging to interpret (**see the following figure**). Secondly, the photooxidation of 2D platelets assembled from **1** is conclusively linked to singlet oxygen. Specifically, singlet oxygen was significantly generated in the solution of 2D platelets assembled from **1** under UV irradiation, as validated by electron paramagnetic resonance spectroscopy (Figure S14), aligning with their swift

photooxidation process (Figure S10). In contrast, 2D platelets assembled from **2** under the same conditions exhibited a significantly lower level of singlet oxygen generation (Figure S14), consistent with their notably slower photooxidation rates (Figure S10). The results presented here are in agreement with prior observations that demonstrate the conversion of fluorene with two alkyl groups to fluorenone through singlet oxygen photooxidation (Adv. Mater. 2009, 21, 597-602). The correlation between photooxidation and singlet oxygen was reinforced by the observation of a minimal presence of singlet oxygen in a toluene solution of molecules **1** and **2** (Figure S15), consistent with their proven high photostability, as illustrated in Figure S10. The significant production of singlet oxygen in the presence of 2D structures, as opposed to the minimal generation of singlet oxygen in solution, emphasizes the crucial role of interactions among the benzoselenadiazole groups in molecule **1**. The intermolecular interactions in 2D structures, demonstrated by the red-shifted charge-transfer (CT) band in comparison to its isolated form (see Figure S16), may promote intersystem crossing (ISC) to the triplet state, possibly by enhancing the heavy atom effect. This, in turn, triggers the generation of singlet oxygen from molecular oxygen. We have added the new data as **Figure S10-12 and S14-16** in the revised supporting information and added more discussion and description in the revised MS to clarify the photooxidation mechanism.

¹H NMR of the platelets fabricated from molecule **1** after 10 min of UV irradiation (365 nm, 300

mW/cm²).

2. A detailed study of the chemistry happening upon UV irradiation must be conducted with other control molecules. The NMR technique may be used to demonstrate the formation of the oxidized species. I wonder what happens when the UV irradiation time increases.

Response: We have compared the photooxidation of molecules **1** and **2** in solution, which exhibited photostable under identical UV irradiation. We have also monitored the level of singlet oxygen produced in the solution, where a minimal amount of singlet oxygen was observed. The observations support that the photooxidation of 2D platelets assembled from **1** result from singlet oxygen (also see the above responses). In addition, we have investigated the photooxidation of another molecule in solution and particles (Adv. Sci. 2021, 8, 2002615), which has been demonstrated to be photostable. As expected, a minimal amount of singlet oxygen was observed under the same conditions. The photostability was further supported by the analysis of photooxidation in solutions containing depicted in Figure S10, consistent with their high photostability.

We have employed MALDI-TOF-MS, a commonly utilized method for identifying various products, to analyze the products resulting from UV irradiation of 2D platelets assembled from **1**. The corresponding data has been included as Figure S11 in the updated supplementary information. Figure S12 illustrates a series of step-by-step oxidized products, with the one containing four fluorenone moieties appearing to be the final product. It is noteworthy that, in contrast to the MALDI-TOF-MS results, the NMR spectrum of the multiple products presented convoluted data that proved challenging to interpret. Also see the above response to comment **1**.

3. Selected area electron diffraction characterizations have not shed any light on the specification of unit cell parameters. The difference in the SAED pattern may be compared before and after the hollow structure formation.

Response: We thank the reviewer for his/her comments. To complement SAED characterizations, we have added the XRD analyses (**Figure S8**), which, together with SAED patterns, can give the unit cell parameters as shown in **Figure S7** in the revised supporting information. Altogether, these data demonstrate that the same molecular packing mode of molecule **1** and **2** in the 2D structures. The optical characterization, including UV-vis absorption and fluorescence spectra, offers macroscopic insights compared to SAED patterns. By comparing the two spectra of segments from **2** before

and after the formation of the hollow structure, it was observed that there was minimal alteration.

4. There is no clear explanation of what the driving force is for forming such a precise, controlled hexagonal 2D structure on a quartz slice in a few seconds.

Response: We thank the reviewer for her/his comment. We have performed the XRD analyses of the resulting 2D structures on a quartz slide, as shown in **Figure S8** in the revised supporting information. The XRD pattern well match the single crystal grown in solution (Figure S8). Therefore, analogous to the bulk crystal (Chem. Eur. J. 2023, 29, e202301747), the electrostatic attraction together with π -interactions lead to the formation of hexagonal 2D structure. We have added the related description in the revised MS.

5. Why is there a discrepancy between the platelet thickness (95 nm) and the thickness (25 nm) of all the electrodes? The extent of sputtering will probably define the thickness of the gold deposition, which is independent of the uniform platelet thickness. From the platelet thickness of 95 nm, it seems like they are not monolayered. How is it possible to control the pi-stacking distance to obtain the uniform thickness of the 2D surface?

Response: Yes, the extent of sputtering defines the thickness of the gold deposition, which is independent of the uniform platelet thickness. Yes, they are not monolayered along the platelet thickness. Referring to the single crystal structure, the molecular packing along the thickness is driven by the interactions between the polar methoxyphenyl groups. This dipole-dipole interactions can be largely weakened by the competitive interaction of polar methoxyphenyl groups with acetonitrile, which thereby suppresses the epitaxial growth along the thickness. Thus, the thickness of 2D platelets can be regulated by manipulating the ratio of the chloroform/acetonitrile mixture.

REVIEWERS' COMMENTS

Reviewer #1 (Remarks to the Author):

The authors have addressed the concerns raised in the first round of review in great detail. The manuscript is greatly improved and thus suitable for publication in Nature Communications at this time.

Reviewer #2 (Remarks to the Author):

This looks good to go now.

Reviewer #3 (Remarks to the Author):

The revised manuscript addressed most of the comments of the reviewers. It may be accepted in the present form.